# Gentle and fast all-atom model refinement to cryo-EM densities via a maximum likelihood approach

**Christian Blau[1], Linnea Yvonnesdotter[2], Erik Lindahl[1,2]\***

**1** Department of Applied Physics, Science for Life Laboratory, KTH Royal Institute of Technology, Stockholm, Sweden, **2** Department of Biochemistry and Biophysics, Science for Life Laboratory, Stockholm University, Stockholm, Sweden

\* erik.lindahl@dbb.su.se

## Abstract

Better detectors and automated data collection have generated a flood of high-resolution cryo-EM maps, which in turn has renewed interest in improving methods for determining structure models corresponding to these maps. However, automatically fitting atoms to densities becomes difficult as their resolution increases and the refinement potential has a vast number of local minima. In practice, the problem becomes even more complex when one also wants to achieve a balance between a good fit of atom positions to the map, while also establishing good stereochemistry or allowing protein secondary structure to change during fitting. Here, we present a solution to this challenge using a maximum likelihood approach by formulating the problem as identifying the structure most likely to have produced the observed density map. This allows us to derive new types of smooth refinement potential—based on relative entropy—in combination with a novel adaptive force scaling algorithm to allow balancing of force-field and density-based potentials. In a low-noise scenario, as expected from modern cryo-EM data, the relative-entropy based refinement potential outperforms alternatives, and the adaptive force scaling appears to aid all existing refinement potentials. The method is available as a component in the GROMACS molecular simulation toolkit.

## Author summary

Cryo-electron microscopy has gone through a revolution and now regularly produces data with 2Å resolution. However, this data comes in the shape of density maps, and fitting atomic coordinates into these maps can be a labor-intensive and challenging problem. This is particularly valid when there are multiple conformations, flexible regions, or parts of the structure with lower resolution. In many cases it is also desirable to to understand how a molecule moves between such conformations. This can be addressed with molecular dynamics simulations using densities as target restraints, but the refinement potentials commonly used can distort protein structure or get stuck in local minima when the cryo-EM map has high resolution. This work derives new refinement potentials based on models of the cryo-EM scattering process that provide a gentle way to fit protein

**Data Availability Statement:** Simulation starting structures, generated densities, setup parameters, complete work- flow setups via Makefiles and Python scripts to generate Fig 2 as well as data for Fig 3 and per-residue RMSD are available via

Zenodo (https://doi.org/10.5281/zenodo.4556616).
The code to perform density-guided molecular
dynamics simulations is maintained within
GROMACS and publicly available in release 2021
and later, as well as in the repository at https://
gitlab.com/gromacs/gromacs. Fourier shell
correlation analysis of tra- jectories has been
implemented on top of the GROMACS codebase
following conventions in EMAN2 [25] and is
available at https://gitlab.com/gromacs/gromacs/-/
commits/fscavg. Python scripts to generate Fig 2,
as well as data for Fig 3 and per-residue RMSD are
available via Zenodo (https://doi.org/10.5281/
zenodo.4556616).

**Funding:** Swedish Research Council (EL; 2017-
04641, 2018-06479, 2019-02433), The BioExcel
Center of Excellence (EL; EU 823830), Knut and
Alice Wallenberg Foundation (EL; 1484505), Carl
Trygger Foundation (EL; CTS-15:298) and the
Swedish e-Science Research Centre. The funders
had no role in study design, data collection and
analysis, decision to publish, or preparation of the
manuscript.

**Competing interests:** The authors have declared
that no competing interests exist.

structures to densities in simulations, and we also suggest an automated heuristic way to balance the influence of the map and simulation force field.

# 1 Introduction

Cryo-electron microscopy (cryo-EM) has undergone a revolution the last few years due to better detectors, measurement techniques and algorithms [1], and the technique now allows for rapid reconstruction of biomolecular "density maps" at near-atomic resolution [2, 3]. These density maps describe interactions between the sample and an electron beam in real space. They (including similar ones derived from X-ray structure factors) provide the basis for reasoning in structural biology. In particular, for cryo-EM, Bayesian statistics has revolutionized the reconstruction of the density maps from micrographs. This provides a framework to soundly combine prior assumptions about the three-dimensional density map model with the likelihood function that connects this model to the measured data and determine the density map most likely to have generated the observed data instead of directly trying to solve the underdetermined inverse problem [4].

However, to understand the structure and function of biological macromolecules, merely having an overall cryo-EM density is typically not sufficient—it is also necessary to model coordinates of individual atoms into the maps [5]. This enables understanding of e.g. binding site properties, interactions with lipids or other subunits, structural rearrangements between alternative conformations, and in particular it makes it possible to model structural dynamics on nanosecond to microsecond time scales via molecular dynamics (MD) simulation [6]. If the interaction descriptions (force fields) used in these simulations were perfect and one had access to infinite amounts of sampling, computational methods should be able to further improve the structure just by starting refinement from a rough initial density, but in practice both force fields and sampling have shortcomings. Nevertheless, it remains an attractive idea to combine the best of both worlds by using cryo-EM data as large-scale constraints while force fields are employed to fine-tune details—in particular details such as local stereogeometry or interactions on resolution scales that go beyond what the cryo-EM data can resolve. Cryo-EM data and stereochemical constraints have been combined favorably in the past to aid structure modeling into three-dimensional cryo-EM densities either by adding force field terms enforcing desired stereochemistry to established modeling tools [7–11] or by adding a heuristic density-based biasing potential to molecular dynamics simulations [12, 13] or elastic network models [14].

In practice, it is not straightforward how to best combine experimental map data with simulations and achieve both satisfactory results and rapid convergence. Density-based biasing potentials can in principle achieve arbitrarily good fits to a map, but it comes at a cost of distorting the protein structure. To address the challenges of balancing desired stereochemical properties with cryo-EM data, refinement protocols have been expanded to include secondary structure restraints [12], multiple resolution ranges [11, 15], as well as multiple force constants [11, 16]; the latter two either consecutively in individual simulations [11] or via Hamiltonian replica exchange [15, 16]. A common challenge of all these approaches is the increase in the ruggedness of the applied bias potential function as the resolution of the cryo-EM density maps increases, and how to correctly balance molecular mechanics and forces from the biasing potential. This leads to an apparent modeling paradox that further improving structural models for cryo-EM densities with molecular dynamics appears to be harder the more high-resolution data is available for these models.

To attack this challenge from a fundamental standpoint, Bayes approach has been used to derive probabilities for all-atom structural models given a cryo-EM density [17] and to weigh cryo-EM data influence against other sources of data [18]. These modeling approaches offer valuable insight into the data content in cryo-EM maps and provide promising new ways to model cryo-EM densities by treating them as generic experimental data. However, they do not reflect the underlying physics of data acquisition and density reconstruction from micrographs and have previously not yielded refinement potentials of a new quality to be applicable, e.g., in molecular dynamics simulations.

One way of circumventing the number of model assumptions that are necessary to reflect the reconstruction of three-dimensional cryo-EM densities is to employ Bayesian models that directly connect micrographs and all-atom ensembles [19]. These attempts have previously proven to be prohibitively costly as a way to derive driving forces for molecular dynamic simulation because projections of model atom coordinates onto millions of cryo-EM particle images (i.e., images of molecules) are required for a single force evaluation.

In this work, we show how it is possible to borrow the highly successful approach to density reconstruction and use maximum likelihood modeling of cryo-EM density maps from structures to derive a new biasing potential that is smooth, long-ranged, and provides fewer barriers to refinement than established potentials based on cross-correlation [11, 13] or inner product (equivalent to using potentials proportional to inverted cryo-EM density [12]). This provides a number of advantages, including an ability to overcome density barriers and in particular avoid excessive biasing forces resulting from large local gradients in cryo-EM density maps. It also avoids the need for constraints e.g. on secondary structure and rather allows the simulation to freely explore local conformational space, while the experimental data is used to bias sampling to experimentally favored regions of the global conformational space.

We further demonstrate how better balancing between the force field and cryo-EM density components can be achieved by adaptive force scaling derived from thermodynamic principles. This allows refinement with a single fixed parameter at low computational cost for a range of system sizes and initial model qualities. Additionally, to evaluate biasing potentials based on model to cryo-EM density comparison, we suggest a transformation of all-atom structure to model density that reflects cryo-EM specific characteristics while minimizing computational effort.

We investigate advantages and drawbacks of the newly derived potential in practical applications when compared to established inner-product and cross-correlation biasing potentials in a noise-free and experimental cryo-EM data. Finally, we show how the proposed refinement methods rectify a distorted initial model with cryo-EM data. A full open-source implementation is freely available as part of the GROMACS molecular dynamics simulation code [20].

## 2 Results

### 2.1 Deriving refinement forces

Our algorithm to refine all-atom models into a cryo-EM density map $\rho$ with molecular dynamics is based on initially roughly aligning density map and target structure, generating a model density from coordinates $\vec{x}$, and then determining a fitting potential $U_{\mathrm{fit}}$ based on a comparison of the generated model density and the target cryo-EM density (Fig 1). This potential is used to derive fitting forces, which are then combined with the force field potential $U_{\mathrm{ff}}$ based on a heuristic balance between the density-derived forces and force field determined by a force constant $k$.

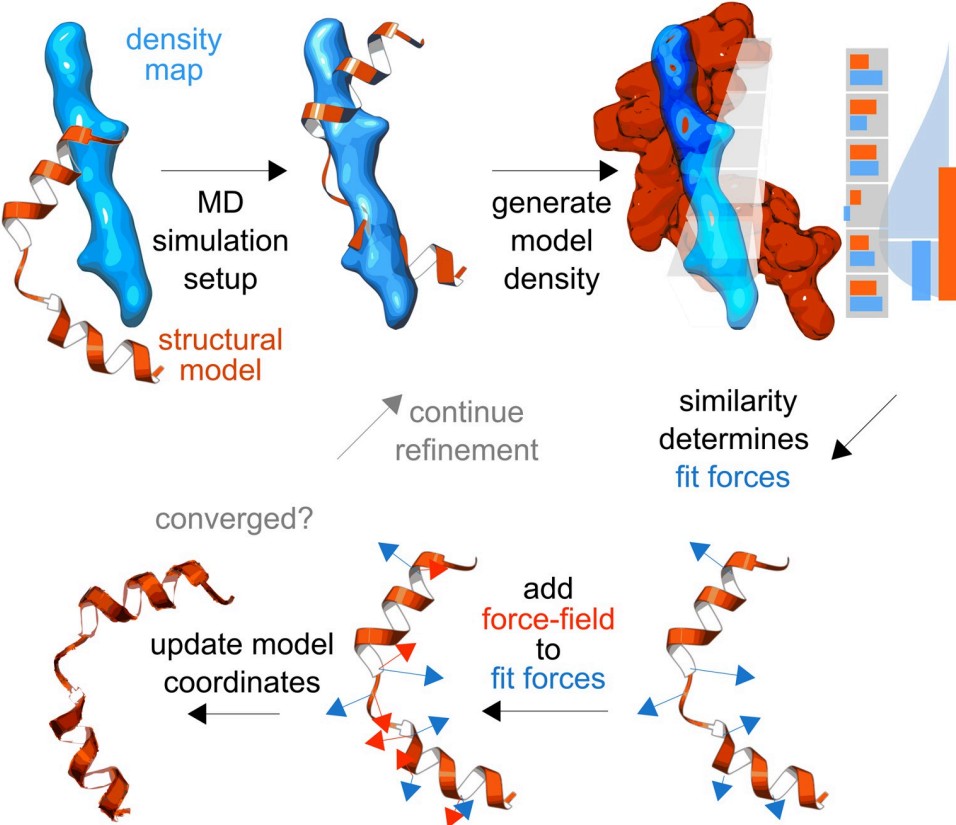

**Fig 1. Atomic structure models are refined into a cryo-EM density using biasing forces that maximize similarity between model and map.** A refinement/simulation is initialized with an atomic model (orange) and a density map (blue). A model density is generated in each voxel (grey boxes). Voxel-wise similarity scores between model density and cryo-EM density are akin to a noise model (light blue curve). The gradient of the similarity score determines the fitting forces (blue arrows). Together with a molecular dynamics force field (red arrows), the fitting forces enable model coordinate updates (dark orange) that make the model more similar to the density under force field constraints. New model densities are generated iteratively from the updated model in each time step of the simulation until acceptable convergence is reached.

The combined driving forces are determined by the total potential energy,

$$U_{\text{tot}}(\vec{x}, \rho) = U_{\text{ff}}(\vec{x}) + U_{\text{fit}}(\vec{x}, \rho) \,. \tag{1}$$

Assuming that a single configuration of atoms gives rise to the observed cryo-EM density, Bayes' theorem quantifies the probability density that the given model describes the cryo-EM data as [17]

$$p(\vec{x}|\rho) \propto p(\vec{x})p(\rho|\vec{x}) \,. \tag{2}$$

Boltzmann inversion at temperature $T$ connects the left-hand sides of Eqs (1) and (2) [21] where $c$ is an arbitrary potential energy offset

$$\log p(\vec{x}) = -\frac{1}{k_B T} U_{\text{ff}}(\vec{x}) + c \tag{3}$$

$$\log p(\rho|\vec{x}) = \frac{1}{k_B T} U_{\text{fit}}(\rho, \vec{x}) + c . \tag{4}$$

In this formalism, the force field provides the prior $p(\vec{x})$ that would have determined the model without any additional observations, while the fitting potential provides the conditional probability that a particular given structure yields a target density $p(\rho|\vec{x})$.

Splitting the fitting potential into a coordinate and density independent force-constant $k$ and a similarity score $S(\rho, \vec{x}) := -U_{\text{fit}}/k$ reveals the similarity to previous approaches [8, 12]. We find that applied density forces imply a similarity score between the model structure and target density, and vice-versa. With this ansatz, Eq (4) relates similarity measures like cross-correlation or inner-product to their implicit assumptions about the likelihood function above, and in turn enables the construction of new similarity scores that drive refinement procedures depending on the assumptions about the underlying measurement process.

## 2.2 Maximum likelihood model yields negative relative entropy as similarity score

To derive a new refinement potential from the likelihood of measuring a density given the structure, we assume that cryo-EM densities are linked to atom-electron scattering probabilities, where electron-atom interaction leads to a phase shift in electrons. With this assumption, two steps are necessary to calculate the density likelihood $p(\rho, \vec{x})$ from coordinates. First, an electron-scattering probability density $\rho^s$ is created from a given structure. Second, this density is compared to a given measured density. Two further assumptions enable the derivation of new similarity scores.

For the per-voxel scattering probability, we present two different sets of assumptions, leading each to a refinement potential in their own right. Many more assumptions may be laid out; here we choose to present those that go beyond previous modeling, yet are well-treatable in model complexity and integratable into a molecular dynamics framework where forces have to be calculated numerically stable and fast. Therefore, we choose not to integrate the full image formation from the microscope detector to three-dimensional density but rather start the modeling process with a three-dimensional density and some additional assumptions on what each voxel represents. By presenting two different approaches to what voxel values in a density present, it should become even clearer how to adapt our modeling to more complex models.

In the first model we assume that an incident electron will scatter at exactly one voxel and that the probability at which voxel to scatter is proportional to the density values. This requires $\sum_v \rho = 1$, which we achieve via rescaling the density values. The free rescaling of cryo-EM density values is motivated from the normalization to unity variance around the particle region during image processing [22]. Introducing the expected number of density interactions, $s$, the scattering process is described via a Dirichlet distribution,

$$p_I(\rho|\vec{x}, s) = \frac{\Gamma\left(\sum_v s\rho_v^s\right)}{\prod_v \Gamma(s\rho_s^v)} \prod_v \rho_v^{s\rho_v^s - 1} . \tag{5}$$

With some approximations (S1 Appendix), we find

$$\log p_I(\rho|\vec{x}, s) = s \sum_{v \in \text{voxels}} \rho_v^s \log \rho_v. \tag{6}$$

This newfound potential relates to the traditionally used refinement potentials by defining the similarity score

$$S_I(\rho, \rho^s) := - \sum_{v \in \text{voxels}} \rho_v^s \log \rho_v . \tag{7}$$

Using this definition we observe that the scaling parameter $s$ and force constant are related via $k = k_B T s$.

Even though $s$ can be estimated from a Bayes' approach with a conditional posterior estimate on $s$ with a prior $p(s)$, to obtain the likelihood to observe a density, given coordinates $\vec{x}$ (see S1 Appendix), we choose a different approach for the following reason. Contributions to this conditional posterior are exponentially weighted with $S_I$, so that a good estimate for the parameter depends on the ability to create a sufficiently representative distribution of $\vec{x}$ similar to the structure. In most cases, however, we expect that we need density-guided simulations to force molecules away from initial configurations over significant energy barriers to be able to sample the relevant configurations that contribute to the estimate of $s$. To overcome these issues, we heuristically scale $s$ with a protocol described below. This allows us to generate a trajectory with structures with ever-increasing likelihood of good structure-to-map overlap, but admittedly at the cost of not sampling from a proper posterior distribution.

In this Dirichlet distribution-based picture the reported density is treated as a probability density, requiring the removal of negative values and normalization to unity. The resulting potential of this modeling approach is proportional to the Kullback-Leibler divergence between simulated and experimental density with a free scaling parameter. This potential in turn can be seen as an inner-product based potential where density is replaced by its logarithm.

In an alternative picture, reported cryo-EM densities at each voxel are proportional to interaction counts $r\rho_v$ of $N$ incident electrons, where the scaling factor $r$ is undetermined as mentioned above. We assume that measured scattering probabilities per voxel are independent of other voxel values. This does not exclude spatial correlation between density data but states that the *scattering process* in one voxel does not influence the electron interaction in other voxels. With this result, it suffices to define a probability distribution at each voxel.

This picture assumes that vitreous ice does not contribute to the scattering, which is commonly achieved by shifting the offset of cryo-EM densities so that water density is represented with voxel values that fluctuate around zero. Only accounting for positive density, we describe this scattering interaction process by a Poisson distribution with parameter $\lambda = N\rho_v^s$. While it is theoretically possible to expand the model to include noise fluctuations and negative densities, we omit this for the sake of reducing model complexity.

With these assumptions (detailed algebraic transformations in S1 Appendix),

$$\log p_{II}(\rho|\vec{x}) = \sum_{v \in \text{voxels}} r\rho_v \log \left(\frac{\rho_v^s}{r\rho_v}\right). \tag{8}$$

In analogy to above, the unknown scaling factor $r$ would be treated as a scale parameter and may be estimated from a conditional posterior distribution, but is left as a free parameter to be heuristically scaled here. We obtain a similarity score between simulated model density and cryo-EM density proportional to the negative relative entropy, or Kullback-Leibler divergence

after normalizing $\rho$ such that $\sum_v r_1\rho_v = 1$, with $r = r_1 r_2$,

$$S_{II}(\rho, \rho^s) := -\sum_{v \in \text{voxels}} r_1\rho_v \log\left(\frac{\rho_v^s}{r_1\rho_v}\right). \qquad (9)$$

Similarly to above, we find that with this similarity score definition, force constant and $r_2$ are related by $k = r_2 k_B T$ (see S1 Appendix).

The newly derived relative-entropy-based similarity score has a domain of $[-\infty, 0]$ with perfect agreement at zero. Due to the $\log \rho_v^s$ term, it differs prominently from established similarity scores like cross-correlation [11] and inner-product (formulated as a force following the gradient of a smoothed inverted density which is equivalent in this approach [12]; see S1 Appendix). In contrast, the relative-entropy based score receives the largest contribution from voxels where cryo-EM data has no corresponding model density data.

This leads to a different behavior from established similarity scores with local minima for locally good agreement with cryo-EM data while the relative-entropy based potential will only have minima where there is good *global* agreement between structure and density. As a consequence, the relative-entropy based density potential is expected to perform better in situations where other potentials cannot escape local minima, at the cost of higher sensitivity to noise in the data, and especially additional density data that is not accounted for in the atomic model.

## 2.3 The potential energy landscapes based on relative entropy are smooth

The proposed relative entropy density-to-density similarity measure has favorable properties in one-dimensional model refinement of one and two particles to a reference density (Fig 2). Both newly derived potentials have the same analytical form for our one-dimensional model case.

In contrast to cross-correlation and inner product similarity measures that have a steep and sudden onset for refinement forces in one dimension, the relative-entropy similarity score has a harmonic shape with long-ranged interactions that allow for efficient minimization. Using relative-entropy, the particle to be refined is attracted by a harmonic spring-like potential to the best-fitting position; far away from the minimum forces are large, but their magnitude decreases monotonically as the minimum is approached. Inner-product and cross-correlation based fitting potentials, however, exert almost no force on the particle outside the Gaussian spread width, while exerting a suddenly increasing force when moving closer to the Gaussian center, and are only insignificant very close to the minimum.

For the refinement of two particles, this advantage is only maintained for one of the newly derived potentials, where the relative-entropy-based potential energy landscape is less rugged and has fewer pronounced features and minima than the corresponding landscapes for the inner-product and cross-correlation based potentials (Fig 2). Only a single diagonal barrier is found in the relative-entropy-based potential landscape, corresponding to a "swapping" of particle positions, which alleviates the search for a global minimum. The inner-product-based free energy landscape has its minimum at a configuration where both particles are at the same position at the highest density. This issue can be alleviated in practical applications through a force-field prior that would enforce a minimum distance between the atoms (e.g. through van der Waals interactions). The swapped relative entropy potential on the other hand exhibits behavior similar to the inner-product, with similar minima but an overall smoother energy landscape.

To model the influence of a force field, the two particles were connected with a harmonic bond with increasing influence. The balance between density-based forces and bond strongly determines the shape of the resulting energy landscape, but here too relative entropy provides a smoother landscape less sensitive to the specific relative weight of refinement and bond potentials.

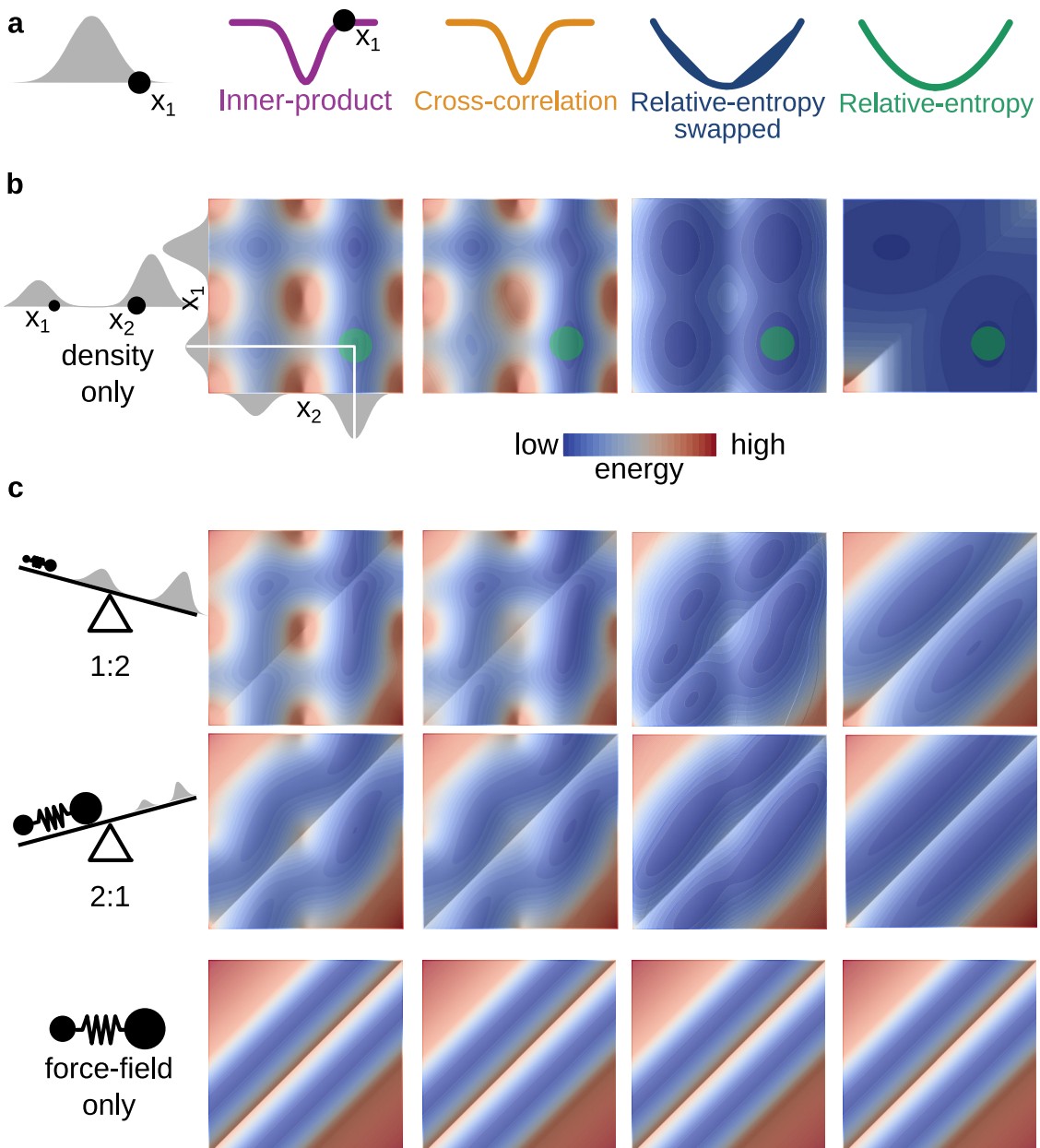

**Fig 2. Similarity score determines ruggedness of the effective refinement potential energy landscape, also when balancing it with structural bias.** From top to bottom: **a** One-dimensional refinement of a single particle (black circle) towards a Gaussian-shaped density (gray) with inner-product (purple), cross-correlation (ochre), relative-entropy swapped (dark blue) and relative-entropy (green) as similarity scores. **b** Expanded model with two particles (black circles, $x_1$ smaller and $x_2$ larger) with two amplitude peaks in a one-dimensional density and target distribution (gray), and the resulting two-dimensional effective potential energy landscapes for inner-product, cross-correlation, swapped relative-entropy and relative-entropy similarity measures. **c** Combination of the similarity measure and force field contribution to the potential energy landscape, exemplified by a harmonic bond that keeps particles at half the distance between the Gaussian centers. For all relative weights of the contributions of the refinement potential and bond potential energy landscape (ratio 1:2 upper panel, 2:1 middle panel, as illustrated by the scale on the left), the relative entropy similarity score produces smooth landscapes with minima at the positions that are expected from the model input.

## 2.4 Adaptive force scaling reduces work exerted during refinement and allows for comparison of density-based potentials

To enable convergence to high similarity between structure and densities without distorting secondary structure, we employ an adaptive force scaling heuristic. Established protocols where the force constant has to be determined manually require an iterative trial-and-error approach. We address this by introducing an adaptive force-scaling as depicted in Fig 3a to automatically balance force-field and density-based forces during the refinement.

Cryo-EM refinement simulations are non-equilibrium simulations with the aim to drive a system from an initial model state to a final state that is as similar to the cryo-EM density as possible while avoiding structural distortions that result e.g. from unphysical paths. To avoid or at least reduce the latter during refinement, a heuristic protocol has been devised that aims

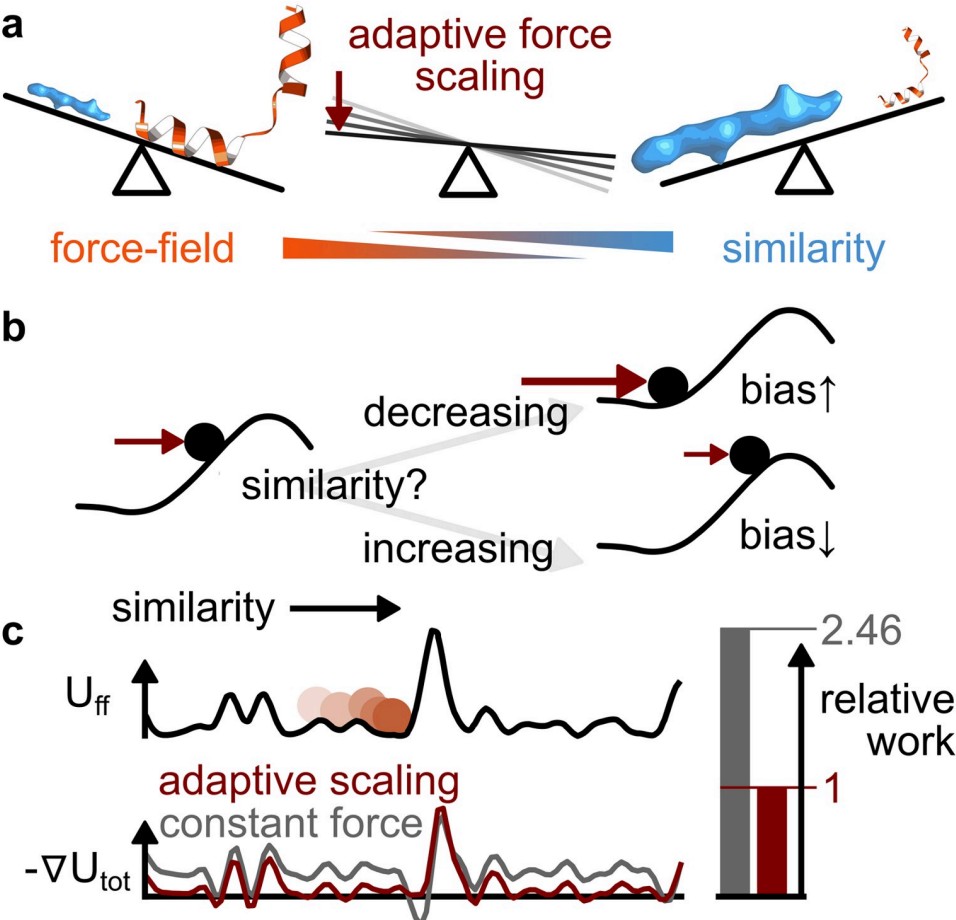

**Fig 3. Adaptive scaling of contributions from force-field and cryo-EM density data overcomes potential energy barriers without excessive work input. a** Adaptive force scaling heuristically balances force-field and density influence during refinement simulations. **b** Particle in energy landscape where density similarity increases from left to right along the black curve. For the upper leg alternative, the similarity decreases despite biasing forces (burgundy arrow), which causes the bias strength to be increased. Conversely, in a scenario where the similarity remains high (lower leg), the biasing force will gradually be reduced to allow the system to better sample the local landscape. **c** Brownian diffusion in a potential with fixed (grey) and adaptive (burgundy) biasing forces, respectively. The constant biasing force is scaled such that both force-adding schemes yield the same average mean first passage time moving from left to right. The relative-entropy approach leads to significantly lower exerted work on the system (area under the grey and burgundy curves, respectively), which reduces perturbation of the dynamics of the system.

to minimize work exerted on the system while still requiring as little time as possible for the refinement.

With similarity $S$ as reaction coordinate when driving the system from an initial fit $S_{\text{start}}$ to $S_{\text{end}}$, the exerted work from the density-guided simulation potential is only determined by the variation of the force-constant as the system progresses:

$$W_{\text{fit}} = \int_{S=S_{\text{start}}}^{S_{\text{end}}} -\partial_S(-kS)\,\mathrm{d}S = \int_{S=S_{\text{start}}}^{S_{\text{end}}} k\,\mathrm{d}S \approx \sum_{\text{frame}} k_{\text{frame}} \Delta S_{\text{frame}}\,, \tag{10}$$

Eq (10) shows that any protocol that decreases $k$ if the $\Delta S > 0$ and conversely increases $k$ if $\Delta S < 0$ decreases is guaranteed to exert less work to reach the same similarity score than keeping the force constant fixed at the final value of the adaptive scaling protocol, given $S_{\text{start}} < S_{\text{end}}$. Fig 3b shows how adaptive-force scaling implements this behaviour, starting from a low force constant $k$.

A one-dimensional Brownian diffusion model system (Fig 3c) is used to test the performance of the concrete scaling protocol as described in the methods section of this paper. In this model, the similarity score simply increases with increasing particle coordinate value. Biasing the system towards increasing coordinate values with adaptive force scaling in contrast to a constant force allows for the particle to reach a given coordinate value in the same average first-passage time at a much lower average work input. Without any coupling of the free energy landscape to the adaptive force scaling protocol other than through the particle trajectory, the adaptive force scaling increases the force just sufficiently to allow overcoming energy barriers but then reduces it again.

Instead of setting a limit to the adaptive force scaling heuristic, we chose to continue the force-scaling until the limits of the molecular dynamics integration algorithm are reached, with the benefit that trajectories are created with structures that represent a wide range of balances between stereo-chemistry and data.

Adaptive force scaling further enables the comparison of relative entropy to other established density-based potentials in simulations with cryo-EM data because it disentangles the effect of the force constant choice from the choice of refinement potential. To carry out this comparison on cryo-EM data with our newly derived similarity score within our framework, a model density generation protocol is required which is shown below.

## 2.5 Deriving an optimal model density generation for cryo-EM data refinement

To evaluate similarities between structural models and cryo-EM densities, a model electron scattering probability density is generated from atom positions. Two dominant effects are convoluted when modeling electron scattering probabilities: The scattering cross-section of each atom and their thermal motion. Both are approximated with Gaussian functions of amplitude $A$ and width $\sigma$. The scattering cross sections determine $A$ (S1 Table). For convenience, we approximate scattering amplitudes by unity for all heavy atoms and zero for hydrogens. The magnitude of thermal fluctuation of atoms at cryogenic temperatures determines the spread width $\sigma$.

In practice, these limitations to the model resolution are superseded by the finite performance of the measurement instrument and the reconstruction process where structural heterogeneity, detector pixel size, microscope lenses, and particle alignment limit the resolution. We do not account for structural heterogeneity, because it is an ensemble effect. A connection between the approach presented here and an ensemble model may be made though by employing a probability distribution $p(\vec{x})$ instead of $\vec{x}$ in Eq (2) and leveraging ensemble

simulations [23]. Other resolution-limiting effects are approximated by additional convolution of the generated maps with a Gaussian kernel. Rather than aiming to reproduce the same blur as in the experimental map, we strive to preserve as much information as possible from the physical model.

A balance between information loss due to under-sampling on the grid on the one hand and information loss due to coarse blurring is found where the Gaussian width at half maximum height equals the resolution. The maximum representable resolution on a grid corresponds to twice the Nyqvist frequency $\delta$ (corresponding to the pixel and voxel size) so that the Gaussian width $\sigma$ is approximated in refinement simulations from the highest local resolution or, where that data is not applicable, from the voxel-size,

$$\sigma = \frac{\text{res}_{\text{max}}}{2\sqrt{\log 4}} \approx \frac{2\delta}{2\sqrt{\log 4}}. \tag{11}$$

For computational efficiency Gaussian spreading is truncated at $4\sigma$ for all simulations in this publication, accounting for more than 99.8% of the density (Fig A in S1 Appendix). The small limitation on the maximal distance between the initial model structure and the cryo-EM density through this cutoff has proven to be irrelevant for all practical purposes, as density-based forces will "pull" structures into densities as soon as there is minimal overlap between model density and cryo-EM density, which can easily be achieved with an initial alignment. Interestingly, this approach results in a smaller Gaussian spreading width than previously applied ones that aim to reproduce a density map with the same overall resolution as the experimental cryo-EM density. As a result, it maintains as much structural information as possible in the model density while still reducing the computational costs.

## 2.6 Refinement against noise-free data

To separate additional noise effects in experimental data and possible limitations in the above model, we first assess refinement with ideal data where a small straight helix model system [14] has been refined against a synthetically generated target density of the same helix in a kinked configuration. As illustrated in Fig 4a, adaptive force scaling and relative-entropy as similarity score efficiently fit the helix into the synthetic cryo-EM density [24].

The combination of adaptive force scaling and different similarity scores achieved a consistent global fit when the helix was aligned to the density, with some fluctuations of the results (Fig 3c) due to the stochastic nature of molecular dynamics simulations. All four similarity measures lead to qualitatively similar evolution of the adaptive force constant (S1 Fig; the absolute value has no meaning since it is merely a relative factor describing the balance between force field and density fitting). Simulations starting from both the aligned and unaligned starting positions occasionally get stuck in local minima, which can result in bad fits. The average total RMSD of all replicates was lowest for relative-entropy starting from the aligned position and further improved when the helix was initially unaligned (S2 and S3 Tables). The relative-entropy based potential shows markedly better results for the unaligned refinement and achieved a fit with less 1Å RMSD in 6 out of 7 replicates while inner-product, cross-correlation, and relative-entropy swapped based potentials in some instances completely fail to align the helix (S2 and S3 Figs). For a single helix this is a slightly artificial case, but in a large structure undergoing significant transitions, it will be common for some secondary structure elements to not overlap with the target density in an initial phase of the refinement.

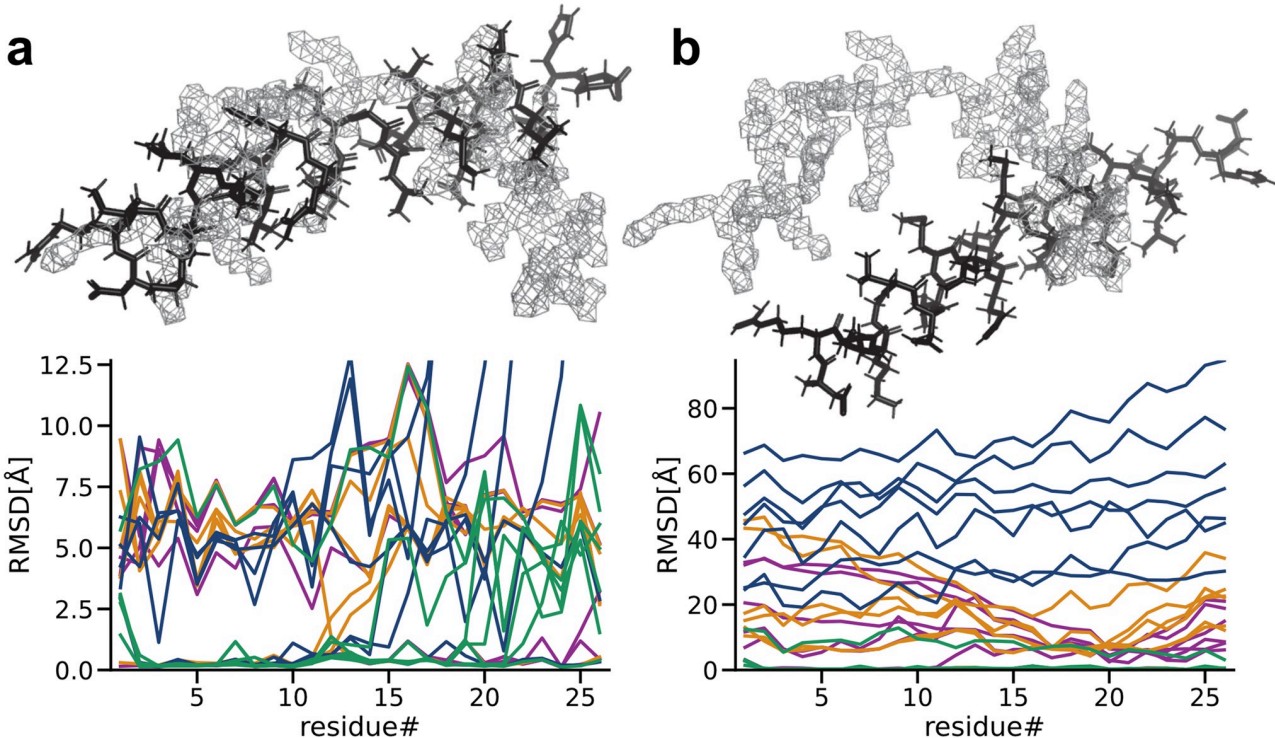

**Fig 4. Refinement into noise-free data with adaptive force scaling. a** Aligned (left) and unaligned (right) starting conformations (black sticks) of a helix subject to refinement simulation into a synthetically generated cryo-EM density (gray mesh). **b** RMSD per residue of the final refined models starting from the aligned conformation compared to the ground truth model underlying the synthetic density map. Each replicate (n=7) is colored by the similarity measure used, inner-product (purple), cross-correlation (ochre), relative-entropy swapped (dark blue), and relative-entropy (green). **c** RMSD per residue of the final refined models starting from the unaligned conformation compared to the ground truth model underlying the synthetic density map. Each replicate (n=7) is colored by the similarity measure used, inner-product (purple), cross-correlation (ochre), relative-entropy swapped (dark blue), and relative-entropy (green).

## 2.7 Refinement against experimental cryo-EM data

Experimental cryo-EM densities of aldolase and a GroEl subunit were used to test the performance of adaptive-force scaling in combination with different refinement potentials on experimental data with increasing amounts of density that is not accounted for in our model description and noise that cannot be fully accounted for by the model assumptions.

By using adaptive-force scaling refinement of a previously published X-Ray structure of rabbit-muscle-aldolase [25] against a recently published independently determined cryo-EM structure [26], we consistently achieve accurate refinement throughout all potentials with good stereochemistry (S4 and S5 Tables). Fig 5a shows the final models of refinement with a global deviation of less than 1Å heavy-atom root mean square deviation (RMSD) from the deposited model using inner-product and cross-correlation measures and just above 1Å for the relative-entropy based density potential.

The close agreement with the cryo-EM data is reflected in the FSC of the models refined against the density (Fig 5b) being nearly indistinguishable from the deposited model. The relative-entropy-based potential emphasizes agreement with global features at the cost of local resolution (S4 Fig), while still providing good agreement to the cryo-EM density.

The unweighted FSC average [27] serves as an established similarity score that is not related to the biasing potentials which were used to refine the system (Fig 5b). All underlying potentials appear to lead to refinement simulations that converge in less than 2 ns, as shown in Fig

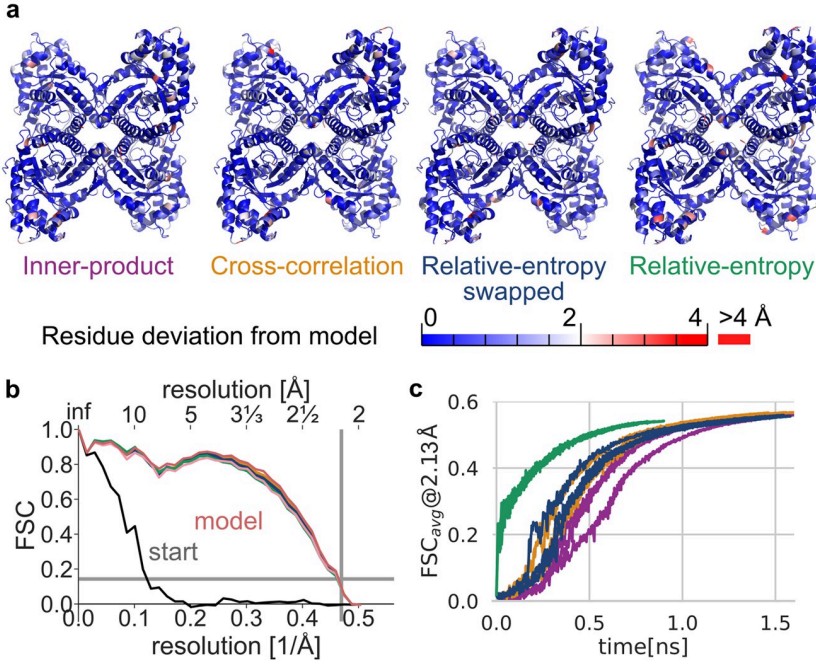

**Fig 5. Refinement of an all-atom X-Ray aldolase structure (PDB id 6ALD) into experimental cryo-EM density (EMD-21023). a** Final structure models from density-guided simulations using different similarity scores colored by unaligned root mean square coordinate deviation (RMSD) per residue from the manually built model (PDB id 6V20). **b** Fourier shell correlation of starting structure (gray line), rigid-body fit of the starting model to the target density (blue) as well as refinement results in the last simulation frame (solid lines). The reported cryo-EM map resolution and 0.143 FSC are indicated with grey lines. **c** Unweighted FSC average over the course of refinement simulation.

5c. The less rugged and long-range potential properties of relative-entropy based density forces are reflected in a rapid rigid-body like initial fit to global structural features, while the other potentials show gradual improvements in fit.

The refinement of a GroEl subunit in two different conformations as determined by cryo-EM [28] stretches the limits of the model assumptions of our refinement potential by refining it against a more noisy model with imperfect map-to-model correspondence. Similar to aldolase refinement, adaptive force-scaling allows for rapid and reliable refinement into the model density, as shown in Fig 6 (S6 and S7 Tables). However, the relative-entropy based potential's propensity to taking all density into account leads to deviations from the published model in regions with density that has no correspondence in the all-atom model (S5 and S6 Figs).

## 2.8 Model rectification by combining force-field and cryo-EM data

To assess performance in larger structural transitions, we repeated the aldolase refinement when starting from initial model structures that have been distorted by heating with partially unfolded secondary structure elements (Fig 7, as described in Methods). Fig 7b shows the final relative-entropy based model of the refinement procedure that achieved 1.13 Å heavy-atom RMSD from the manually built model. Structural details at map resolution match in secondary structure elements. In contrast to refinement of the undistorted X-ray structure, the relative-entropy based potential gains less from the long-rangeness of the potential and the rapid alignment of large-scale features, because structural rearrangements were needed on all length scales. The adaptive force scaling protocol alleviates differences between density-based

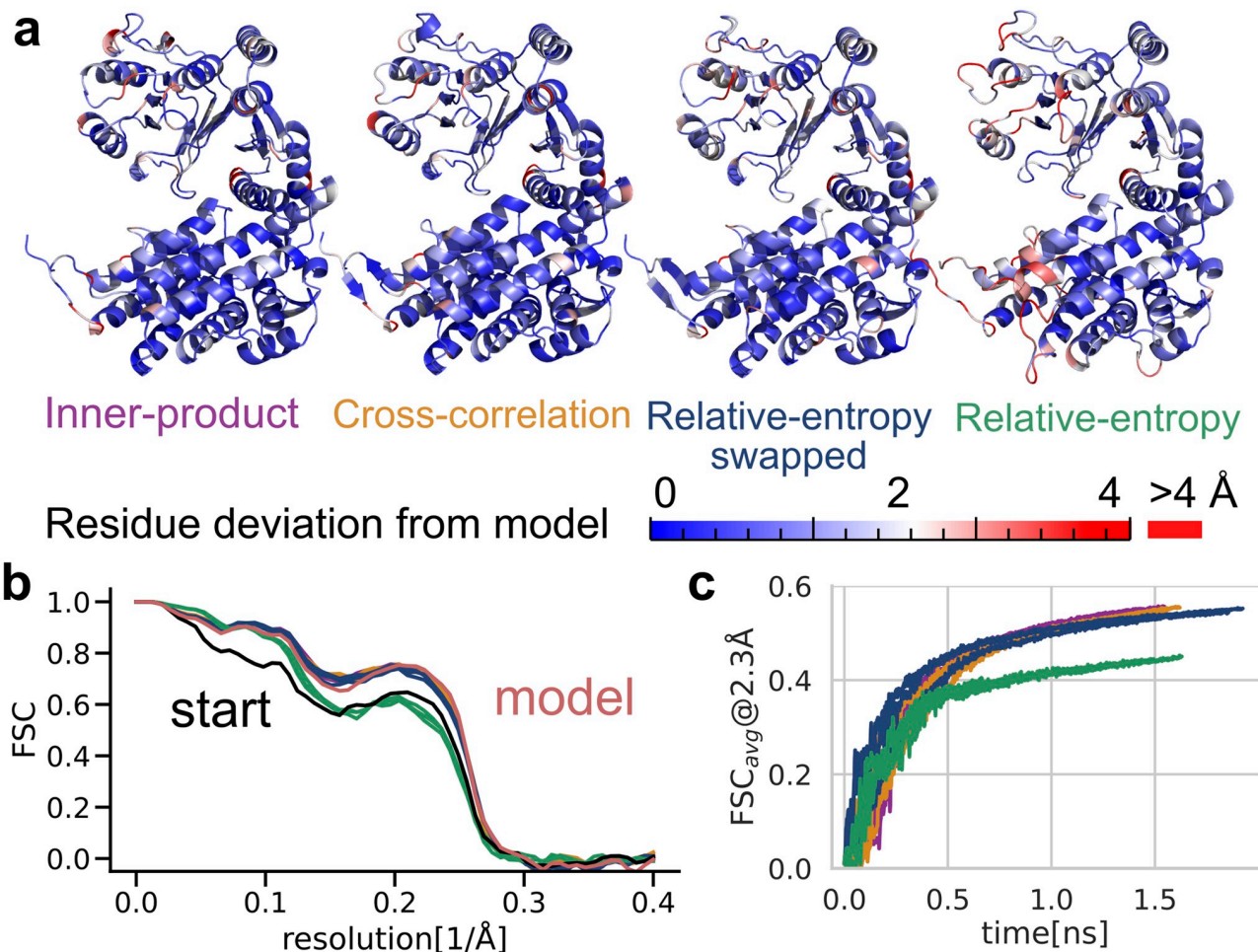

**Fig 6. Refinement of an all-atom GroEL cryo-EM structure (PDB id 5W0S state 1) into experimental cryo-EM density (EMD-8750, additional map 3). a** Final structure models from density-guided simulations using different similarity scores colored by unaligned root mean square coordinate deviation (RMSD) per residue from the deposited model (PDB id 5W0S state 3). **b** Fourier shell correlation of starting structure (gray line), rigid-body fit of the starting model to the target density (blue) as well as refinement results in the last simulation frame (solid lines) deviations of an equilibrium simulation (dotted lines). **c** Unweighted FSC average over the course of refinement simulation.

potentials in refinement speed and allows for refinement with good structural agreement in less than 3 ns (S7 Fig). The adaptive force scaling protocol allows the modeling to be more steered by cryo-EM data and reach model structures that would not have been accessible by modeling using stereochemical information from the force-field alone.

## 3 Discussion

While defining a purely empirical similarity measure can sometimes suffice to fit structures to cryo-EM densities, connecting the similarity measure to the underlying measurement process of the target density enables derivation of natural similarity measures. From very few assumptions, this results in density-based potentials derived from the maximum likelihood that coincides with the relative-entropy between a model density generated from model/simulation atom coordinates and the target cryo-EM density.

The newly defined potentials have favorable features, with the Poisson statistics-based relative entropy potential most prominently exhibiting long range and low ruggedness. It avoids

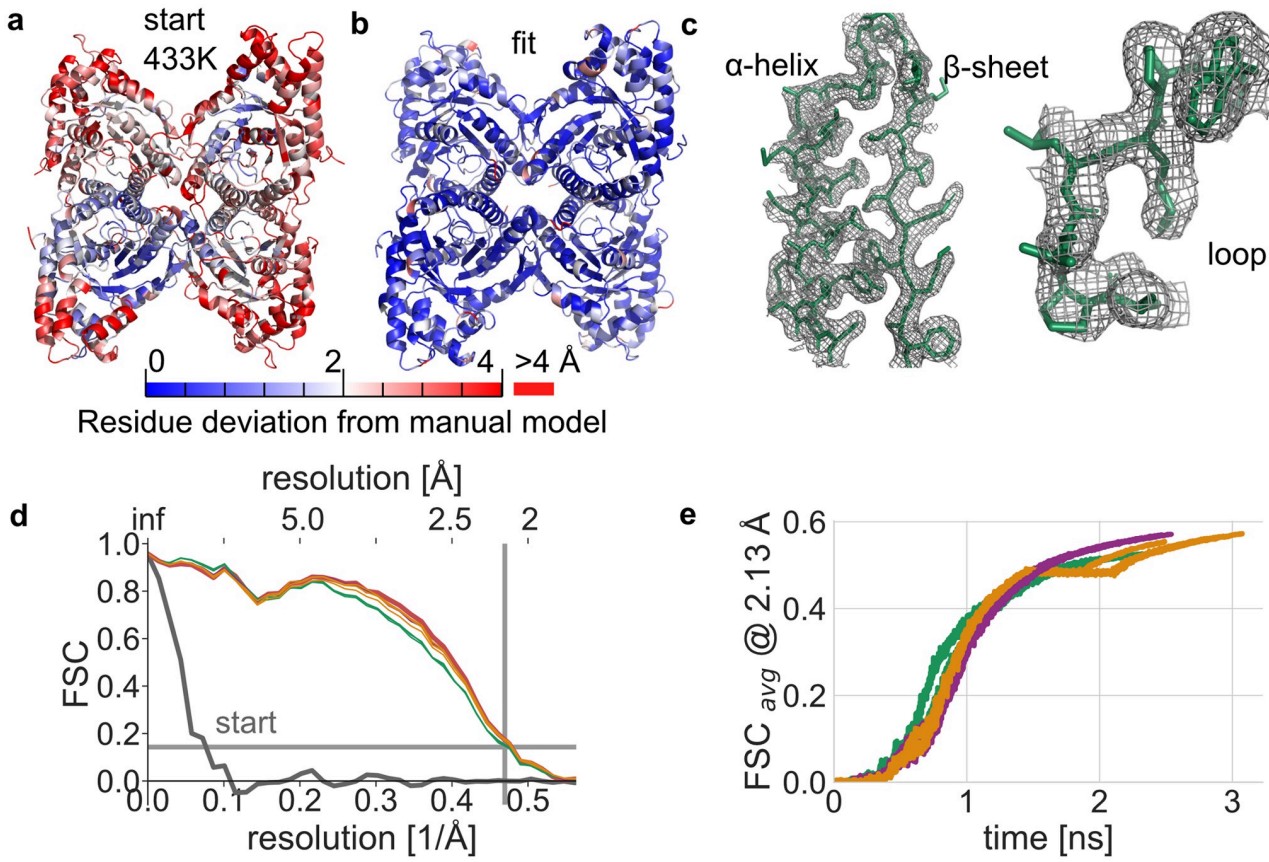

**Fig 7. Cryo-EM data rectifies model distortions with density-guided simulations. a** Distorted starting model RMSD with respect to manually built model (PDB id 6V20). **b** Final model structure after refinement into a cryo-EM density (EMD-21023) using adaptive force scaling and relative-entropy similarity score. **c** Close-up of structural features of the final simulation model (green lines) and cryo-EM density (gray mesh). **d** Fourier shell correlation of starting structure (gray line) as well as refinement results in the last simulation frame (solid lines). The reported cryo-EM map resolution and 0.143 FSC value are indicated with grey lines. **e** Un-weighted FSC average over the course of a refinement simulation.

local minima that do not correspond to desired configurations during refinement and allows rapid alignment of large-scale features, and performs superior to established refinement in the zero noise setting with synthetic density maps. The noise content in current cryo-EM densities is likely still too high to be handled with the current minimalistic model assumptions, but as the quality of cryo-EM and other low-to-medium resolution techniques continues to rapidly improve, we believe there will be even more advantages to models that do not depend on smoothing. In addition, the adaptive force-scaling provides a surprisingly simple way to tackle the inaccessibility of the balance between force-field and density-based forces within our model assumptions. It allows for parameter-free refinement that is one to two orders of magnitude faster than currently established protocols. Conceptually it is orthogonal to, and easily combined with, multi-resolution protocols [15].

Another illustration of the usefulness of our framework for handling force-field vs. fitting forces is how it enables us to deduce a close-to-optimal model density spread for refinement, and even more so that this value is not identical to the common practice of setting it equal to the experimental resolution. While many of these factors could still be tuned manually, removing them as free parameters means fewer arbitrary settings that avoid over- or underfitting, which will be even more important when trying to combine e.g. multiple sources of

experimental data. For trial structure refinement against recent cryo-EM data, we show that we achieve excellent fits independent of initial model quality.

One limitation of the current formulation is that it does not explicitly take more information from the cryo-EM density reconstruction process into account. A first step to broaden the approach presented here is to account for the local resolution information from the cryo-EM map, which may be seen as a measure of the noisiness of the density data. In practice, the local resolution will still influence the fit since low local resolution will correspond to smoother regions of the map, and lower-magnitude gradients will lead to lower-magnitude fitting forces in those regions. However, a formally more correct way to address the problem is likely to treat the target cryo-EM density as a statistical distribution with a variance that is spatially resolved—this is something we intend to pursue in the future to see whether it can further improve the issue of the relative-entropy potential with more noisy data.

Our current algorithm opens the path to a fully Bayesian approach by generating structures from which the correct probability distribution and the scale parameters $s$ and $r$ for the Dirichlet and Poisson model, respectively, may be estimated and thus estimate the correct balance between force-field and cryo-EM data. This will also work as an alleviation of the short-coming that the adaptive-force scaling is an inherently non-equilibrium approach, which might thought of as an "initial structure generator" for further equilibrium sampling.

The algorithms proposed in this work are freely available, integrated, and maintained as part of GROMACS [29]. Overall, three independent building blocks are provided to aid the modeling of cryo-EM data that each may be individually implemented in current modeling tools: A new refinement potential, a new criterion for how to calculate the model density, both based on reasoning from a maximum likelihood approach, and adaptive force scaling to gently and automatically bias stereochemistry and cryo-em data influence. The implementation also provides tools to monitor the refinement process. Although it can still be difficult for any automated method to compete with manual model building by an experienced structural biologist, we believe these methods provide new ways to extract as much structural information as possible from cryo-EM densities at minimal human and computational cost, which is particularly attractive e.g. for fully automated model building.

## 4 Methods

### 4.1 Calculating density-based forces

For ease of implementation and computational efficiency the derivative of Eq (4) is decomposed into a similarity measure derivative and a simulated density model derivative, summed over all density voxels $v$

$$\mathbf{F}_{\text{density}} = k \sum_{v} \partial_{\rho_v^s} S(\rho, \rho^s) \cdot \nabla_{\mathbf{r}} \rho_v^s(\mathbf{r}) . \tag{12}$$

Though the convolution in eq (12) might be evaluated with possible performance benefits in Fourier space, we have implemented the more straightforward real-space approach.

The forward model $\rho^s$ is calculated using fast Gaussian spreading as used in [30]; the integral over the three-dimensional Gaussian function over a voxel is approximated by its function value at the voxel center $\mathbf{v}$ at little information loss (Fig B in S1 Appendix). Amplitudes of the Gaussian functions [31] have been approximated with unity for all atoms except hydrogen. The explicit terms that follow for $S(\rho, \rho^s)$ and $\nabla_{\mathbf{r}} \rho_v^s(\mathbf{r})$ are stated in the S1 Appendix.

## 4.2 Multiple time-stepping for density-based forces

For computational efficiency, density-based forces are applied only every $N_{fit}$ steps. The applied force is scaled by $N_{fit}$ to approximate the same effective Hamiltonian as when applying the forces every step while maintaining time-reversibility and energy conservation [32, 33]. The maximal time step should not exceed the fastest oscillation period of any atom within the map potential divided by $\pi$. This oscillation period depends on the choice of reference density, the similarity measure, and the force constant and has thus been estimated heuristically.

## 4.3 Adaptive force scaling

Adaptive force constant scaling decreases the force-constant when similarity increases by a factor $1 + \alpha$, with $\alpha > 0$, and reversely increases it by a factor $1 + 2\alpha$ when similarity decreases. The larger increase than decrease factor enforces an increase in similarity over time.

To avoid spurious fast changes in force-constant, similarity decrease and increase are determined by comparing similarity scores of an exponential moving average. The simulation time scale is coupled to the adaptive force scaling protocol by setting $\alpha = \frac{N_{fit}\Delta t}{\tau}$, where $\Delta t$ is the smallest time increment step of the simulation and $\tau$ determines the time-scale of the coupling.

This adaptive force scaling protocol ensures a growing influence of the density data in the course of the simulation, eventually dominating the force-field. Simulations with adaptive force scaling are terminated when overall forces on the system are too large to be compatible with the integration time step.

## 4.4 Comparing refined structures to manually built models

Root mean square deviations (RMSD) of all heavy atom coordinates (excluding hydrogen atoms) used absolute positions without super-position as structures because the cryo-EM density provides the absolute frame of reference. This is an upper bound to RMSD values between refined and manually built models calculated with rotational and translational alignment.

## 4.5 Comparing refined structures to cryo-EM densities

Fourier shell correlation curves and un-weighted Fourier shell correlation averages [27] were calculated at 4 ps intervals from structures during the trajectories by generating densities from the model structures using a Gaussian $\sigma$ of 0.45 Å, corresponding to a resolution as defined in EMAN2 [24] to 2Å.

## 4.6 Map and model preparation before refinement

**4.6.1 Helix.**   Noise-free helix density maps at 2Å simulated resolution on a 1 Å voxel grid were generated from an atomic model using "molmap" as provided by chimera [34]. Two frames taken from an equilibration simulation of the helix model were used as starting models for subsequent density-guided refinement.

**4.6.2 Aldolase.**   A simulation box of the exact dimensions as aldolase density map EMD-21023 was used. The corresponding aldolase model (PDB id 6V20) was treated as a ground truth for RMSD calculations. A previously determined X-ray model (PDB id 6ALD) was used as starting structure. The system was subjected to energy minimization before fitting. No symmetry constraints were used in simulations.

**4.6.3 GroEL.**   The density-guided simulations of GroEL were performed between conformational states within an individual oligomer [28]. State 1 (PDB id 5W0S(1)) was used as starting model and fit toward conformational state 3 (EMD-8750, additional map 3). A section of the density map corresponding to a single oligomer was used as the target. The target map was

created using ChimeraX [35] and by including any volume data within a 5 Å range of the previously determined model corresponding to state 3 (PDB id 5W0S(3)). The target density was centered in a map and the map size was set to $100^3$ using relion image handler [36].

### 4.7 Generation of a distorted model

To generate a distorted starting model, the aldolase protein X-ray was heated to $433\frac{1}{3}$ K over a period of 5 ns. During heating, the pressure was controlled with the Berendsen barostat, favoring simulation stability over thermodynamic considerations. To disentangle effects from decreasing the temperature and fitting, the distorted structure was subjected to 5 ns of equilibration at 300 K before starting the density-guided simulations with the same protocol as described above.

### 4.8 Molecular dynamics simulation

All simulations were carried out with GROMACS version 2021.3 [29] and the CHARMM27 force-field [37, 38] in an NPT and NVT ensembles with neutralized all-atom systems in 150 mM NaCl solution. The temperature was regulated with the velocity-rescaling thermostat at a coupling frequency of 0.2 ps to ensure rapid dissipation of excess energy from density-based potential, when structures are very dissimilar from the cryo-EM density, i.e., far from equilibrium. The pressure was controlled with the Parinello-Rahman barostat for aldolase simulations, helix simulations were carried out at constant volume. Aldolase and GroEL were aligned roughly to the density by placing their center of geometry in the center of the cryo-EM density box. Forces from density-guided simulations were applied every $N_{fit} = 10$ steps according to the protocol described above [33]. All simulations to refine a structure against a density were carried out with adaptive force-scaling. We used a coupling constant of $\tau = 4$ ps, balancing time to result with time for structures to relax. For aldolase simulations, the Gaussian spread width was determined by using a lower bound on the highest estimated local resolution of 1.83 Å. Spread width for GroEl simulations was set to 1.0455Å, based on the 1.23Å voxel size in the map used for refinement. Periodic boundary conditions are treated as described in the S1 Appendix. All simulation setup parameters and workflows have been made available.

## Supporting information

**S1 Appendix. Derivation of refinement potentials.** Derivation of relative entropy measures from Poisson noise assumption, grid scattering assumptions, model density gradients and similarity measure definitions.
(PDF)

**S1 Table. Scattering cross sections.** Scattering amplitudes for common atoms in biological molecules at 150keV, derived from Appendix C in Ref. [31].
(PDF)

**S2 Table. Aligned helix fitting.** Heavy-atom RMSD [Å] at the final frame compared to conformation from which density was generated.
(PDF)

**S3 Table. Unaligned helix fitting.** Heavy-atom RMSD [Å] at the final frame compared to conformation from which density was generated.
(PDF)

**S4 Table. Aldolase fitting.** Heavy-atom RMSD [Å] from final simulation frames, as compared to PDB id 6V20.
(PDF)

**S5 Table. Aldolase model statistics.** Properties calculated with PHENIX-1.18.2–3874, using MOL-PROBITY, CABLAM, and EMRINGER methods. All data reflect the final frame without any further geometry optimization.
(PDF)

**S6 Table. GroEl fitting.** Heavy-atom RMSD [Å] from final simulation frames, as compared to conformational state 3 of PDB id 5W0S.
(PDF)

**S7 Table. GroEl model statistics.** Properties calculated with PHENIX-1.18.2–3874, using MOL-PROBITY, CABLAM, and EMRINGER methods. All data reflect the final frame without any further geometry optimization.
(PDF)

**S1 Fig. Adaptive force constant change.** Evolution of force constant (arbitrary units) during aligned helix refinement for inner-product(purple), cross-correlation(ochre), relative-entropy (green) and relative-entropy-swapped (blue).
(PDF)

**S2 Fig. Final results of aligned helix fitting.** Simulations using adaptive force scaling from aligned starting conformation using inner-product, cross-correlation, relative-entropy swapped, and relative entropy (ordered top to bottom).
(PDF)

**S3 Fig. Final results of unaligned helix fitting.** Simulations using adaptive force scaling from aligned starting conformation using inner-product, cross-correlation, relative-entropy swapped, and relative entropy (ordered top to bottom).
(PDF)

**S4 Fig. Aldolase refinement.** Difference in FSC to deposited Aldolase model for inner-product (purple), cross-correlation (ochre), swapped relative-entropy (blue) and relative-entropy (green) based on refinement final frames (solid).
(PDF)

**S5 Fig. GroEL subunit refinement.** Difference in FSC to deposited GroEL model for inner-product (purple), cross-correlation (ochre), swapped relative-entropy (blue) and relative-entropy (green) based on refinement final frames (solid).
(PDF)

**S6 Fig. Relative entropy can be sensitive to additional density.** Relative entropy-based refinement (blue-red, according to RMSD to published model) can struggles in regions with extra density present in the map, whereas the cross-correlation based potential (ochre) adheres to the local minimum defined by the density.
(TIF)

**S7 Fig. Aldolase refinement from distorted structure.** Difference in FSC to manually built model for inner-product (purple), cross-correlation (ochre), relative-entropy swapped (dark blue) and relative-entropy (green) based refinements with best accepted FSC (dotted) and final FSC (solid).
(PDF)

## Acknowledgments

We would like to thank Rebecca J. Howard and Marta Carroni for insightful discussions of the manuscript.

## Author Contributions

**Conceptualization:** Christian Blau.

**Formal analysis:** Christian Blau, Linnea Yvonnesdotter.

**Funding acquisition:** Erik Lindahl.

**Investigation:** Christian Blau, Linnea Yvonnesdotter.

**Methodology:** Christian Blau.

**Project administration:** Erik Lindahl.

**Resources:** Erik Lindahl.

**Software:** Christian Blau.

**Supervision:** Erik Lindahl.

**Validation:** Christian Blau, Linnea Yvonnesdotter.

**Visualization:** Christian Blau, Linnea Yvonnesdotter, Erik Lindahl.

**Writing – original draft:** Christian Blau, Erik Lindahl.

**Writing – review & editing:** Christian Blau, Linnea Yvonnesdotter, Erik Lindahl.

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
