## [Decision Letter · Decision Letter 0]

1 Nov 2022

Dear %TITLE% Lindahl,

Thank you very much for submitting your manuscript "Gentle and fast all-atom model refinement to cryo-EM densities via Bayes' approach" for consideration at PLOS Computational Biology.

As with all papers reviewed by the journal, your manuscript was reviewed by members of the editorial board and by several independent reviewers. In light of the reviews (below this email), we would like to invite the resubmission of a significantly-revised version that takes into account the reviewers' comments.

Reviewers 1 and 2 are overall positive about the proposed refinement method. They request additional tests and a few clarifications to be able of evaluating the method. Reviewer 3 is more critical and raises conceptual concerns, in particular on the use of the heuristic force constant (as also mentioned by Reviewer 2) and on the assumption of a Poisson likelihood. Reviewer 3 also questions whether using a single experimental test case is sufficient for stating that the method outperforms alternatives.

We cannot make any decision about publication until we have seen the revised manuscript and your response to the reviewers' comments. Your revised manuscript is also likely to be sent to reviewers for further evaluation.

Sincerely,

Jochen Hub

Academic Editor

PLOS Computational Biology

Nir Ben-Tal

Section Editor

PLOS Computational Biology

Reviewers 1 and 2 are overall positive about the proposed refinement method. They request additional tests and a few clarifications to be able of evaluating the method. Reviewer 3 is more critical and raises conceptual concerns, in particular on the use of the heuristic force constant (as also mentioned by Reviewer 2) and on the assumption of a Poisson likelihood. Reviewer 3 also questions whether using a single experimental test case is sufficient for stating that the method outperforms alternatives.

Reviewer's Responses to Questions

**Comments to the Authors:**

Reviewer #1: In this work, the authors proposed to use relative entropy derived from Bayes’ approach in all-atom model refinement to cryo-EM density data. In the model refinement, molecular force field and similarity bias force are both used. The conventional similarity bias forces are inner product, cross correlation, and so on. The proposed relative entropy seems to have advantages over these conventional similarity bias forces because the effect of similarity bias is longer range even though the overlap between structural models and cryo-EM densities is very small. Adaptive force scaling looks also useful for adjusting the good balance between molecular force field and similarity bias forces. In general, this work can provide new insight and useful information in the all-atom model refinement to cryo-EM density data. We would like to suggest several points to improve the manuscript and to clarify the advantages of relative entropy in the refinement.

(1) For noise-free data, is it possible to test different initial structures ? For instance, not only completely unaligned structures but also (one or two residue) miss-aligned ones could be tested.

(2) To compare the performance of inner-product, cross-correlation with relative-entropy, not only FSC but also EMRinger, MolProbity, CABLAM should be shown.

(3) In Figure 6, they used distorted models for showing the performance of relative-entropy approach. We suggest them to show more practical examples for showing the performance. For instance, there are several proteins whose structures were determined with X-ray crystallography and another structures were solved with cryo-EM at the lower resolution. We would like to see how the relative entropy shows good performance to refine all-atom models with lower resolution cryo-EM densities using the X-ray structure at the different physiological states. As we suggested, not only FSC but also EMRinger, MolProbity, and CABLAM could be used to investigate the performance.

Reviewer #2: The manuscript by Blau et al. describes an algorithm to refine protein structures against cryo-EM density maps that is based on molecular dynamics simulations. In their description, the assumption is made, that the EM density map describes a single conformation and the goal of the refinement is a single, best-fitting model.

Applications to simulated and experimental density maps are shown. Overall, this is a very interesting manuscript, which describes a new development of the MD-based structure refinement in Gromacs.

The method seems to be well applicable and fast, it is therefore a valuable contribution for the (computational) structural biology community and well suited for PLOS Computational Biology.

Here are a few points that need to be clarified:

- A new refinement potential is derived, which results in the Kullback-Leibler divergence between model density and EM density. The authors emphasize that this refinement potential is smooth. I would think that ideally, all experimental information should be used during the refinement. I understand, that high-resolution density maps are rugged and lead to slower convergence, since many barriers need to be crossed, but this is often accounted by a stepwise refinement, starting with low-pass filtered maps (as e.g. described in cascade MDFF), and then at later stages of the refinement, the full-resolution maps are used. It does not sound like a desirable feature to use only a smooth refinement potential throughout and thereby possibly ignoring valuable high-resolution information. The authors might want to clarify, whether indeed high-resolution is potentially ignored with their approach.

- Where does the 3sigma potential energy threshold come from? This is not really described in the text.

- In the applications to experimental data, the model was fit against half map1? Why was that done? The half maps are usually the raw output of the density refinement program (e.g. Relion), the maps are typically not filtered and not sharpened and should therefore not be used for structure refinement.

- The FSC threshold for the resolution definition is 0.143 when comparing the two half maps (after a gold-standard refinement). The FSC threshold for comparing a model map with the full EM map (sum of both half maps) (also referred to as the model-map cross-resolution) is 0.5.

That means, if the half map FSC is 0.143 at a resolution value XÅ, the model-map FSC should have a value of 0.5 at the same resolution XÅ. However, in figure 5 and 6 this is not the case.

Here, the model is compared to only a half map, which is unusual and makes this comparison more difficult.

This needs to be clarified.

- page 10: "The force constant for density-guided simulations cannot be derived from the cryo-EM density alone and thus needs to be set heuristically."

In principle, the force constant could in fact be determined from the cryo-EM data, it requires to model the error of the density map. The main motivation to use Bayes (in my opinion) is that it does not require any tunable parameters if all errors are modeled. However, in this work, there are still several tunable parameters: the force-constant, the filter of the density maps, and the 3sigma potential cut-off selects the best structure from the trajectory. The error of the density map could in principle be modeled based on the FSC. When I read "Bayes" in the title, I immediately expected that the force constant would be obtained by modeling the experimental errors. It might be helpful to the reader to clarify, why this was not done.

- The relative-entropy based density potential enforces a global agreement of model and density.

How does that affect the typical practical problem of missing model (e.g. missing long loops, missing domains)? Would you expect larger distortions than with the traditional refinement target functions?

- Throughout the manuscript: Cryo-EM does not yield an "electron density" map, but an electrostatic potential map. Please do not use "electron density". (X-ray diffraction yields electron density)

Appendix S1, "II Gaussian Spreading on a Grid":

"we assume Gaussian spatial noise", I do not understand why this has anything to do with noise?

This is just a convolution with a Gaussian, or not?

And how is the factor A chosen, this should depend on atom type. There is a reference to a textbook by Kirkland, but I think it would be helpful to just list values for the few most relevant atom types in proteins.

typos:

page 7: "vitr*e*ous ice"

page 10: "This issue can *be* alleviated in..."

page 14: "excees" -> "exceeds"

Appendix page 4: "Wi*gn*er-Seitz"

Reviewer #3: see attached review

**Have the authors made all data and (if applicable) computational code underlying the findings in their manuscript fully available?**

Reviewer #1: Yes

Reviewer #2: Yes

Reviewer #3: Yes

PLOS authors have the option to publish the peer review history of their article (what does this mean?). If published, this will include your full peer review and any attached files.

Reviewer #1: No

Reviewer #2: No

Reviewer #3: No
---

## [Decision Letter · Decision Letter 1]

29 Mar 2023

Dear %TITLE% Lindahl,

Thank you very much for submitting your manuscript "Gentle and fast all-atom model refinement to cryo-EM densities via Bayes' approach" for consideration at PLOS Computational Biology.

As with all papers reviewed by the journal, your manuscript was reviewed by members of the editorial board and by several independent reviewers. In light of the reviews (below this email), we would like to invite the resubmission of a significantly-revised version that takes into account the reviewers' comments.

Reviewers 1 and 2 confirmed that their comments have been largely addressed. Reviewer 3, however, points out that several mathematical equations as well as new text added during the revision are not justified or not sufficiently explained. Their criticism on the fact that the method is not "Bayesian" remains (because it is never sampled from the posterior).

From Reviewer 3's report and from my own reading, there is no need for new simulations or analysis prior to acceptance, but the criticism can be solved by revising the discussion and the mathematical derivations and motivations. Most critically, a discussion on how to render the method "Bayesian" in a future study is needed, or to tone down the claim that the method would be Bayesian. The criterium used to stop the simulation must be stated (since Fig. S3 does not reveal any convergence). Apart from following the comments by Reviewer 3, please also revise Eq. 9., as the partial derivate of S with respect to S should be one. Maybe you need to use a different integration variable here?

We cannot make any decision about publication until we have seen the revised manuscript and your response to the reviewers' comments. Your revised manuscript is also likely to be sent to reviewers for further evaluation.

Sincerely,

Jochen Hub

Academic Editor

PLOS Computational Biology

Nir Ben-Tal

Section Editor

PLOS Computational Biology

Reviewers 1 and 2 confirmed that their comments have been largely addressed. Reviewer 3, however, points out that several mathematical equations as well as new text added during the revision are not justified or not sufficiently explained. His criticism on the fact that the method is not "Bayesian" remains (because it is never sampled from the posterior).

From Reviewer 3's report and from my own reading, there is no need for new simulations or analysis prior to acceptance, but the criticism can be solved by revising the discussion and the mathematical derivations and motivations. Most critically, a discussion on how to render the method "Bayesian" in a future study is needed, or to tone down the claim that the method would be Bayesian. The criterium used to stop the simulation must be stated (since Fig. S3 does not reveal any convergence). Apart from following the comments by Reviewer 3, please also revise Eq. 9., as the partial derivate of S with respect to S should be one. Maybe you need to use a different integration variable here?

Reviewer's Responses to Questions

**Comments to the Authors:**

Reviewer #1: The authors replied to almost all the questions and comments raised by three reviewers. We are satisfied about their replies to our previous comments. In particular, Figure 6, a new figure in the revised manuscript, seems to be meaningful to understand the behavior of relative entropy in the refinement.

However, before the publication of this manuscript, we suggest to correct the following points:

(1) add the label "C" in Figure 6. I can find only the labels "A" and "B" in Figure 6.

(2) Probably, the vertical axis "FSC_AVG@2.3A" seems to be wrong. 1/2.3 = 0.434. Figure 6b suggests that FSC value of this resolution (0.434) is almost zero. Please use the correct resolution in the vertical axis.

Reviewer #2: My concerns have been fully addressed in the revised version of the manuscript.

Reviewer #3: see attached PDF

**Have the authors made all data and (if applicable) computational code underlying the findings in their manuscript fully available?**

Reviewer #1: Yes

Reviewer #2: Yes

Reviewer #3: Yes

PLOS authors have the option to publish the peer review history of their article (what does this mean?). If published, this will include your full peer review and any attached files.

Reviewer #1: No

Reviewer #2: No

Reviewer #3: No
---

## [Editor Report · Decision Letter 2]

9 Jun 2023

Dear %TITLE% Lindahl,

Thank you for further improving the manuscript. The power of the method together with remaining room for future developments are now clearly stated. Congratulations to this insightful work!

We are pleased to inform you that your manuscript 'Gentle and fast all-atom model refinement to cryo-EM densities via a maximum likelihood approach' has been provisionally accepted for publication in PLOS Computational Biology.

Best regards,

Jochen Hub

Academic Editor

PLOS Computational Biology

Nir Ben-Tal

Section Editor

PLOS Computational Biology

---

## [Editor Report · Acceptance letter]

23 Jul 2023

PCOMPBIOL-D-22-01441R2 

Gentle and fast all-atom model refinement to cryo-EM densities via a maximum likelihood approach

Dear Dr Lindahl,

I am pleased to inform you that your manuscript has been formally accepted for publication in PLOS Computational Biology. Your manuscript is now with our production department and you will be notified of the publication date in due course.

With kind regards,

Bernadett Koltai
